# Seasonal Variability of Photosynthetic Microbial Eukaryotes (<3 µm) in the Kara Sea Revealed by 18S rDNA Metabarcoding of Sediment Trap Fluxes

**DOI:** 10.3390/plants10112394

**Published:** 2021-11-06

**Authors:** Tatiana A. Belevich, Irina A. Milyutina, Aleksey V. Troitsky

**Affiliations:** 1Biological Faculty, Lomonosov Moscow State University, 119234 Moscow, Russia; 2Belozersky Institute of Physico-Chemical Biology, Lomonosov Moscow State University, 119992 Moscow, Russia; iramilyutina@yandex.ru

**Keywords:** microbial eukaryotes, photosynthetic picoeukaryotes, sediment trap, seasonal succession, NGS, the Kara Sea

## Abstract

This survey is the first to explore the seasonal cycle of microbial eukaryote diversity (<3 µm) using the NGS method and a 10-month sediment trap (2018–2019). The long-term trap was deployed from September to June in the northwestern part of the Kara Sea. A water sample collected before the sediment trap was deployed and also analyzed. The taxonomic composition of microbial eukaryotes in the water sample significantly differed from sediment trap samples, characterized by a high abundance of Ciliophora reads and low abundance of Fungi while trap samples contained an order of magnitude less Ciliophora sequences and high contribution of Fungi. Photosynthetic eukaryotes (PEs) accounting for about 34% of total protists reads were assigned to five major divisions: Chlorophyta, Cryptophyta, Dinoflagellata, Haptophyta, and Ochrophyta. The domination of phototrophic algae was revealed in late autumn. Mamiellophyceae and Trebouxiophyceae were the predominant PEs in mostly all of the studied seasons. *Micromonas polaris* was constantly present throughout the September–June period in the PE community. The obtained results determine the seasonal dynamics of picoplankton in order to improve our understanding of their role in polar ecosystems.

## 1. Introduction

Microbial photosynthetic eukaryotes (PEs) with cell diameters <3 µm are one of the major components of phytoplankton in the Arctic region. They are responsible for significant photosynthetic production and therefore have a significant impact on marine carbon and energy budgets [1,2,3]. In the Arctic Ocean, picoeukaryotes account for 36% of the autotrophic biomass from June to August [4]. In the Canadian Arctic in late summer, the estimated chlorophyll a biomass of cells <2 μm was responsible for 1% to 96% of the total chlorophyll a along the transect [5]. In the Russian Arctic seas, the contribution of photosynthetic picoeukaryotes to the total phytoplankton biomass and primary production in summer averages 10–61% and 41–60%, respectively [6].

In polar regions, microbial eukaryotes are well adapted to extreme environmental conditions [7,8]. Their taxonomic composition depends on environmental parameters such as day length, solar radiation, temperature, salinity, ice cover, and wide variations in river runoff [9,10]. The last two factors, ice cover and river runoff, have greater seasonal variations due to climatic changes that occur twice as fast in the Arctic than at lower latitudes [11]. According to estimates, the changes in the pelagic marine environment occurring in the Arctic region have led to an increase in the proportion of phototrophic picoeukaryotes in the total planktonic algal biomass due to the penetration of moderate waters algae and the disappearance of Arctic endemics [12,13]. Moreover, species with smaller sized cells may adapt faster to changing environmental conditions [14,15] because of smaller plastic genomes and higher reproductive rates following an r-selection strategy.

Sediment traps deployed in different areas of the world ocean provide estimates of particle sinking rates and particle composition in the deeper water column e.g., [16,17,18]. At present, studies of the variability in protist community composition with the application of long-term sediment traps and molecular methods are scarce. Previously, Metfies et al. [19] showed that combining sediment traps and next generation sequencing (NGS) might qualitatively determine pelagic eukaryotic microbial biodiversity. Phototrophic picoeukaryote diversity with the application of molecular methods has been studied in the Arctic region, mostly in spring and summer [7,8,20,21]. To date, knowledge of seasonal changes in picoeukaryote taxonomic composition, including phototrophic eukaryotes, has been limited [10,22]. Russian Arctic seas, including the Kara Sea, are a hard-to-reach and understudied region. The Kara Sea is a typical Arctic shelf sea influenced by the significant freshwater runoff exceeding a volume of 1200 km^3^/year [23,24]. The St. Anna Trough is located in the northwestern area of the Kara Sea and connects the shallow sea shelf (depths up to 100 m) and deep areas of the Arctic Ocean. The desalinated Kara Sea shelf waters, Arctic basin waters, and transformed Atlantic waters create complex hydrophysical and hydrochemical conditions in the St. Anna Trough. The interaction of different waters in the trough affects the community composition of pelagic microbial eukaryotes. Picophytoplankton studies in the Kara Sea have been limited to the ice-free period, which usually lasts from July to December [25,26]. Specifically, studies of picophytoplankton diversity have been carried out only in early (end of August) and late autumn (end of September) [27,28].

A primary objective of this study was to investigate communities of microbial eukaryotes (<3 µm) in the northwestern Kara Sea using fluxes from annual sediment trap and high-throughput metabarcoding of the hypervariable V4 region of the 18S rDNA and describe the seasonal (September–June) diversity of these microbial eukaryotes.

## 2. Materials and Methods

### 2.1. Study Area and Sampling

During the 72 research expedition of the R/V Akademik Mstislav Keldysh (September 2018), a long-term 12-cup conical sediment trap LOTOS-3 with a sampling area of 0.5 m^2^ (the Experimental Design Bureau of Oceanological Engineering, Russian Academy of Sciences) was deployed in the Kara Sea at station 5976; 76°34.3′ N, 71°15.0′ E (Figure 1) on 8 September 2018. The sediment trap was deployed at a water depth of 110 m, approximately 60 m off the bottom. Ten collecting cups were filled with filtered seawater (GF/F 0.7 μm) that was adjusted to a salinity of 40 with NaCl and 1% HgCl_2_ to preserve the samples during deployment and after recovery. The samples were collected at 30-day intervals from September 2018 to June 2019. After trap rising in July 2019 (76 Research Expedition of the R/V Akademik Mstislav Keldysh), the sample cups were refrigerated until further processing.

Since the trap was set on 8 September 2018 and the first cup was changed after 30 days (8 October), we considered the sample from the first collection cup (T1) as the September sample, the sample from the second cup (T2) was considered the October sample, etc. Thus, in our study, all samples from the trap were sequentially named in order from T1 to T10 and corresponded to the period from September 2018 to June 2019. The sample from station 5976 collected before the sediment trap was referred to as water sample 5976.

The abundance and biomass of picophytoplankton, the concentration of the total chlorophyll a (Chl_tot_) and chlorophyll a of the picofraction (Chl_pico_) were determined at station 5976 during the setting of the sediment trap in September 2018. Additionally, to identify the microbial eukaryote diversity at the trap location a 3 L water sample (3 L) was filtered through a 3 µm pore size polycarbonate filter using a <50 mmHg vacuum. The filtrate was then filtered through a 0.2 µm Sterivex unit (Millipore Canada Ltd., Mississauga, ON, Canada). Buffer was added to the Sterivex unit (1.8 mL of 50 mM Tris–HCl, 0.75 M sucrose, and 40 mM EDTA; pH 8.3), and the sample was stored at −80 °C until nucleic acid extraction.

### 2.2. Picophytoplankton Enumeration

We used epifluorescence microscopy (Leica DM2500, Biosystems Switzerland AG, Muttenz, Switzerland) to count picophytoplankton directly on board the research vessel. The sub-samples (10 mL) were placed in a filtration funnel and incubated for 5–7 min after a saturated solution of primulin was added. Each sample was preserved with glutaraldehyde at a final concentration of 1%. Nuclear filters (0.12-μm pore diameter, Dubna, Russia) were prestained with Sudan black and were used for filtration. The cells on the filter were counted under a Leica DM1000 epifluorescence microscope at a ×100 × 10 × 1.3 magnification. Depending on the cell concentration, 30 to 50 fields were examined, and the cell size was measured. The “type” of fluorescence was also determined: spherical cells with a diameter ≤1.5 μm with orange fluorescence from phycoerythrin (575 ± 20 nm) were considered to be picocyanobacterial. Orange fluorescence under blue excitation is also specific to cryptophytes, but the latter can be easily identified by their asymmetric cell shape and were absent in our samples. Cell volume was converted to carbon using different conversion factors. For the picocyanobacteria, in which cell sizes varied from 0.8 to 1.2 µm (average 1 µm), a conversion factor of 470 ƒg C/cell was used [29]. The carbon biomass of the picoeukaryotes was estimated according to a conversion factor of logC = 0.941 logV − 0.60 [30].

### 2.3. Chlorophyll a Measurement

Chlorophyll a concentrations were determined using the fluorometric method [31]. To determine Chl_tot_, seawater samples (500 mL) were filtered over Whatman GF/F glass fibre filters under low vacuum pressure (~0.3 atm) and then extracted in 90% acetone at 5 °C in the dark for 24 h. The fluorescence of the extracts was measured using a fluorometer (Trilogy, Turner Designs, San Jose, CA, USA) before and after acidification with 1 N HCl. The fluorometer was calibrated before and after the cruise using pure chlorophyll *a* (Sigma-Aldrich, St. Louis, MO, USA) as a standard. Chlorophyll a concentration was calculated according to Holm-Hansen and Riemann (1978) [31].

To determine Chl_pico_ concentration, 1000-mL water samples were prefiltered by gentle reverse filtration through nuclear filters with 3-μm pores (JINR, Dubna, Russia). The resulting filtrate was processed as described above.

### 2.4. DNA Isolation

In the laboratory, sample cups were gently shaken to suspend cells in a homogeneous solution before a subsample (10 mL) was taken. Large swimmers were removed using a 1-mm mesh nylon sieve. The subsample from every cup was filtered through a 3 µm pore size polycarbonate filter using a <50 mmHg vacuum. The filtrate was then collected onto a 0.2 μm Millipore Isopore membrane filter (Millipore, Schwalbach, Germany). Filters were washed with sterile Kara Sea water (~50 mL) as recommended by Metfies et al. [19]. Total DNA from the filters and from the Sterivex unit was extracted with a NucleoSpin Plant Kit (Macherey-Nagel, Düren, Germany) following the manufacturer’s instructions.

A subsequent analysis of DNA was carried out according to Belevich et al. [28]. A fragment of the 18S rDNA containing the hypervariable V4 region was amplified with the primers EuF-V4(f) (5′-CCAGCASCCGCGGTAATWCC-3′) and picoR2(r) (5′-AKCCCCYAACTTTCGTTCTTGAT-3′) [32]. The library preparation and sequencing of the DNA fragments were carried out with TruSeq Nano DNA Kit according to the manufacturer’s protocol by using the Illumina MiSeq system (Illumina, San Diego, CA, USA). The read length was 250 bp; reading was performed from both sides of the fragments. The sequencing was conducted by BioSpark (Moscow, Russia).

### 2.5. NGS Read Processing and Phylogenetic Analysis

The amplicon sequence variant (ASV) method, which resolves single-nucleotide differences and provides constant labels across studies, was used. Sequence reads were processed with the DADA2 inference algorithm [33] on primer-free reads to correct sequencing errors and create ASVs for picoplankton communities. The reads were quality filtered, dereplicated, and merged, chimaeras were then removed according to Tutorial R Dada2 metabarcode analysis (https://vaulot.github.io/tutorials/R_dada2_tutorial.html; accessed on 17 August 2019). The ASVs were assigned taxonomically against the Protist Ribosomal Reference database (PR2 version 4.12.0 https://github.com/pr2database/pr2database/releases, (updated 17 August 2019). Moreover, the ASVs related to phototrophic picoeukaryotes also were checked manually against the NCBI nucleotide database using BLAST [34]. All classifications and representative sequences are available at Appendix A. The taxonomy of the PEs was determined by AlgaeBase [35].

### 2.6. Statistical Analyses

The relative abundance of the reads was used for a comparison of the samples. Significance was determined by analysis of variance (ANOVA) tests, Bray–Curtis dissimilarity matrices, and diversity indices were calculated using Primer v6 software [36].

## 3. Results

### 3.1. Environmental Conditions

During the year, the air temperature varied from 2 °C in September 2018 to −29 °C in February 2019 (Figure 2). The air temperature was below zero for eight months out of the 10 months that were monitored. The sediment trap was deployed under ice-free conditions, which were maintained until mid-December 2018 (Figure 2). The sea ice covering the area from November 2018 to early June 2019, began to melt in the middle of June and dropped to 0% in mid-July. During November, December, and January, the sun stayed below the horizon for at least 24 h (the polar night). Accordingly, a polar day occurred in May and June (Figure 2).

Station 5976 is located in the upper part of the St. Anna Trough slope in the northeastern part of the Kara Sea. At the beginning of September, the water temperature of the upper 20-m layer varied from +0.2 to −0.4 °C, and the salinity was 33‰. The depth of the euphotic zone (Z_eu_, 1% of surface irradiance) was 35 m.

### 3.2. Chl_tot_, Chl_pico_, Abundance, and Biomass of Photosynthetic Picoeukaryotes at Station 5976 While Setting the Sediment Trap

The maximum Chl_tot_ was detected in the surface layer (0.43 mg/m^3^) and decreased with depth when the sediment trap was deployed in September (Figure 3a). In comparison to the Chl_tot_, the Chl_pico_ concentration varied less and ranged from 0.13 mg/m^3^ at the surface to 0.12 mg/m^3^ at the 40 m horizon. The contribution of Chl_pico_ to Chl_tot_ increased with the depth from 30% at the surface layer to 40% at a 40 m depth. Picophytoplankton were represented exclusively by eukaryotes, and cyanobacteria were not detected in the studied samples. The maximum values of the abundance and biomass of photosynthetic picoeukaryotes (3.88 × 10^9^ cells/m^3^ and 2.76 mg C/m^3^, respectively) were found in the surface horizon and decreased with depth (Figure 3b).

### 3.3. General Characteristics of Microbial Eukaryote Taxonomic Composition in Samples Filtered through a 3 µm Pore Size Filter

A total of 1,516,007 quality-filtered sequences were obtained from the analysis of the 11 samples. Protist reads accounted for 76% to 99% of the quality-filtered sequences (Table 1). The remaining sequences related to Metazoa (Crustacea, Cnidaria, Annelida, Ctenophora, Hexapoda, and Rotifera) and Streptophyta (Embryophyceae) were excluded from the analysis.

Different ASVs were grouped according to their taxonomic affiliations with major phylogenetic supergroups, such as Alveolata, Hacrobia, Opisthokonta, Rhizaria, Stramenopiles, and Archaeplastida (Figure 4). Protists from tRhizaria and Opisthokonta taxonomic groups are nonphotosynthetic forms. Alveolata, Hacrobia, and Stramenopiles include phototrophic, mixotrophic and heterotrophic species. The relative abundances of Apicomplexa, Perkinsea, Conosa, Centroheliozoa, Mesomycetozoa, and Opalozoa were lower than 1% in every sample. A total of 723 ASVs related to 291 protist forms identified as belonging to groups of different taxonomic ranks were found in all samples. Based on the total number of ASVs analyzed, approximately 90% and 60% of them could be assigned to the class and order levels, respectively. Most genera were represented by several ASVs. Overall, reliable identification at the genus level was possible for 57% of the ASVs (159 genera).

In September, Alveolates were the most prominent taxonomic group of the pelagic eukaryotic community at station 5976 before the sediment trap was installed (Figure 4). The main contributors were Ciliophora and Dinoflagellata, accounting for 38% and 32% of the total protist reads, respectively (Figure 4). A classification of phylotypes at lower taxonomic levels (e.g., genus or species) was difficult for both dinoflagellates and ciliates. Ciliophora were represented by abundant biosphere ASVs (each accounting for >1% of the total relative abundance of reads at the sampling station) relating to different Strombidiida phylotypes, for which relative abundance was 96% of the total ciliate reads.

Dinoflagellates were more diverse than ciliates and were represented mainly by Syndiniales (an average of 86% of all Dinoflagellate reads) consisting of mostly picoplanktonic parasites. Ochrophyta and Chlorophyta accounted for on average 11% and 8% of the total protist reads, respectively.

Three phyla predominated in terms of richness in all the studied trap samples: Dinoflagellata, Fungi, and Chlorophyta. They represented an average of 66% of the total richness and accounted for 72% of the total reads. Dinoflagellata, Fungi, and Chlorophyta collectively accounted for significant contributions in autumn (samples T1, T2, and T3) comprising an average of 14%, 21%, and 43% of the total protist reads, respectively. In addition to the dominant phyla, nearly 19% of the sequences affiliated with the Radiolaria order Chaunacanthida were identified in September (T1). In winter (T4, T5, and T6), Fungi dominated only in December (T4), and their relative contribution reached 42%. In January and February, Dinoflagellata was the most important group, accounting for more than 40% of the total reads. Chlorophyta accounted for an average of 21% of the total sequence reads in these samples. In samples T7, T8 and T9, the average contribution of Dinoflagellata reads reached 27%. Chlorophyta and Fungi accounted for an average of 22% and 19% of the total sequence reads, respectively. In the June sample (T10), 45% of the total reads were affiliated with Chlorophyta. Moreover, 20% of all sequences were affiliated with Radiolaria.

Fungi were identified in all trap samples at high abundances (≥10%), while in the September water sample their relative abundance was near 1%. Most of the fungal reads were assigned to Ascomycota and Basidiomycota, and their relative contributions to the total fungal reads ranged from 97% to 100%. Cryptomycota and Microsporidiomycota were found only in the T6, T7, T9 and T10 samples. In September water sample 5976, only two Basidiomycota genera of yeasts, *Rhodotorula* sp. and *Malassezia* sp., were identified.

Dinoflagellata were represented by three classes: Syndiniophyceae, Oxyrrhidophyceae, and Dinophyceae. Syndiniophyceae were dominant in almost in all samples, with an average contribution of 73% to the total Dinoflagellata reads. In samples T4 (December) and T5 (January), Syndiniophyceae were the only identified group. All Syndiniophyceae ASVs were assigned to the class level. The contribution of Dinophyceae averaged 23%, and varied from 6% in T2 (October) to 55% in T1 and T6 (September and February). Oxyrrhidophyceae were found only in three samples, T1 (September), T7 (March), and T8 (April), and were represented by two ASVs of the genus *Oxyrrhis* sp. All Syndiniophyceae and Oxyrrhidophyceae are heterotrophs, while Dinophyceae also includes mixotrophic and phototrophic algae. Fourteen Dinophyceae ASVs were identified to the genus level or a higher taxonomic level and were related to phylotypes with “ambiguous trophic status” (Figure 5).

The total ASV richness in samples T7–T10 (170 ± 86) appeared more diverse than that in samples T1–T3 (58 ± 10) and T4-T6 (74 ± 31); however, these seasonal differences were not statistically supported (*p* = 0.151).

### 3.4. Variability in Pelagic PE Diversity in the Samples Filtered through a 3 µm Pore Size Filter

The PEs in the trap and water samples accounted for approximately 34% of the total protist reads, with their percentage varying among samples between 16% (T8) and 53% (T2 and T3) (Table 1). The number of PE ASVs clustered in individual samples varied between 13 and 81 (Table 1).

At a higher taxonomic level, the ASVs of the PEs were assigned to five major divisions: Chlorophyta, Cryptophyta, Dinoflagellata, Haptophyta, and Ochrophyta (Figure 6). All identified ASVs were related to sixteen classes of algae and two higher-ranked taxonomic groups.

All studied samples were characterized by high amounts of Chlorophyta, representing 78% per sample on average (ranging from 35% reads in T6 to 93% in T2 and T10), except water sample 5976 and T6, where Ochrophyta and Dinoflagellata, respectively, were the dominant groups (Figure 6).

The community composition of PEs in the water sample 5976 and the sediment trap samples varied significantly. Ochrophyta was the most abundant group in the water sample and was mainly represented by Bacillariophyta (37% of all PE reads), followed by Dictyochophyceae (12% of all PE reads). Ochrophyta were represented by the diatoms *Fragilariopsis cylindrus*, *Skeletonema* sp., *Chaetoceros socialis*, *Synedra hyperborea* and *Thalassiosira hispida*, and by the Dictyochophyceae *Dictyocha speculum* and *Florenciella parvula*. Two ASVs, *F. cylindrus* and *Skeletonema* sp., accounted for 20% and 8% of total PE reads, respectively. In the water sample 5976, all Chlorophyta ASVs were affiliated with the class Mamiellophyceae and consisted of *Micromonas polaris*, *M. commoda*, *Micromonas* clade F according to [37] or B3 [32], *Mantoniella squamata*, *Bathycoccus prasinos* and one Dolichomastigaceae-B phylotype. The first two taxa noted were most abundant at 24% and 6% of total PE reads (Figure 7).

Chlorophyta was dominated by Mamiellophyceae in almost all trap samples and was the most diverse group of PEs (106 ASVs). Only in February (T6) and March (T7) was this algal class in the third and second most abundant groups, respectively (Figure 6). Six Mamiellophyceae species and three taxa of the order Dolichomastigales were observed (Figure 7). *M. polaris* was found in all samples, followed by *Micromonas* clade B3/F in seven out of 10 samples. *M. polaris* and *Micromonas* clade B3/F represented 51% and 3% reads per sample on average, respectively. *M. polaris* dominated in almost all trap samples with the exception of T6 and T7. Other Mamiellophyceae, *M. commoda*, *B. prasinos*, *Ostreococcus tauri* and *M. squamata* together with two phylotypes of Dolichomastigaceae and Crustomastigaceae were completely absent in the autumn and winter samples and appeared in the photosynthetic protist communities only in February or March (L6–L7) with the end of PN. One phylotype, Dolichomastigaceae-B, was abundant and was found in the autumn and the winter samples in six out of 10 samples. According to NCBI, this phylotype corresponds to uncultured eukaryote clone DSGM-81 (AB275081), which was previously identified in methane cold-seep sediment of Sagami Bay.

Trebouxiophyceae was the second Chlorophyta class that was found in all trap samples and accounted for, on average, approximately 17% of the PE reads. *Choricystis* sp. and *Picochlorum* sp. were the most abundant and occured in all seasons, and their greatest contribution to total PE reads was found in samples T4–T7 at an average of 8% and 28%, respectively. Three more Trebouxiophyceae phylotypes, *Neocystis brevis*, *Botryococcus* sp., and *Chlorella* sp., were identified at low abundances in only a few samples. Trebouxiophyceae were not found in water sample 5976.

Chlorophyta diversity was also indicated by algae from Pyramimonadaceae, Palmophyllophyceae, Chlorodendrophyceae, Chlorophyceae, and Chloropicophyceae. A total of 20 ASVs affiliated with 10 species and five genera were identified in the trap samples, mostly in the winter and spring samples, where they represented from 0.2% to 13% of the total PE sequences (Appendix A).

Bacillariophyta was the second most diverse group (50 ASVs); however, it was not abundant in the trap samples but was abundant in water sample 5976. Diatom community composition showed different patterns in the individual samples. *Chaetoceros socialis* was the most common species and found in almost all trap samples. *Ch. socialis* represented from 0.2% to 10% of all PE reads and was followed by *Skeletonema* sp., found in four out of 10 trap samples at low abundances <1% (Figure 7). The highest relative abundance of diatom reads was detected in December (T4) and March (T7), where they reached almost 10% of the PE reads per sample. Bacillariophyta in March (T7) was the most diverse and was represented by 12 phylotypes (16 ASVs). In addition to the above mentioned above *Ch. socialis* and *Skeletonema* sp., *Amphora pediculus*, *Bacillaria paxillifer*, *Ch. gelidus*, *Cocconeis pediculus*, *C. placentula*, *Cyclotella choctawhatcheeana*, *Fragilariopsis cylindrus*, *Gyrosigma acuminatum*, *Melosira arctica*, *Naviculales* sp., *Staurosira construens*, *Synedra hyperborean*, *Thalassiosira anguste-lineata*, *Th. hispida*, *Th. nordenskioeldii*, *Th. rotula*, *Th. tenera* and *Thalassiosira* sp. were found (Appendix A).

Bolidophyceae was represented by two phylotypes and identified in only two samples, water sample 5679 and in the March (T7) sample, at low abundances of 2% and <1%, respectively. Both Bolidophyceae phylotypes were identified to the order Parmales and affiliated with phylotypes previously found in the pico-sized phytoplankton fraction of the White Sea [38].

Four ASVs were assigned to two species of Eustigmatophyceae, *Nannochloropsis granulate* and *Pseudocharaciopsis ovalis*, and one unidentified phylotype, Eustigmatophyceae. The maximum read abundance of Eustigmatophyceae was found in the March sample (T7) and did not exceed 3%.

Pelagophyceae was represented by three phylotypes, *Ankylochrysis* sp., *Aureococcus anophagefferens* and *Pelagococcus* sp. *A. anophagefferens* was identified in the September (T1) trap sample, while the remaining two were found in the March (T7) trap sample.

Dictyochophycea played an important role in the September water sample 5976. The ASVs of *Dictyocha speculum*, *Florenciella parvula* and two more phylotypes affiliated with the families Florenciellales and Pedinellales were identified in the water sample, where their relative abundance exceeded 12% of the total PE reads. In comparison to in the water sample 5976, in the sediment trap, Dictyochophycea were less represented; only Pedinellales were found in two samples (T3 and T7) at low relative abundances <1%.

Phototrophic Cryptophyceae were detected in most trap samples in low proportions (<4% reads per sample on average) but occasionally reached higher proportions, for example, in March (9% reads in T7) and in April (14% reads in T8). Cryptophyceae were identified based on 12 ASVs affiliated with three different species: *Baffinella frigidus*, *Plagioselmis prolonga* and *Rhodomonas abbreviata*. The last species was widespread and identified in all trap samples except T2 and T3. In water sample 5976, phototrophic Cryptophyceae were not found.

Dinophyceae was the only class of Dinoflagellata that included phototrophic algae that were approximately 7% of all Dinoflagellata sequences. The highest proportion of 34% was found in February (L6). Twenty-two ASVs of phototrophic Dinophyceae were identified in the studied samples. Generally, the contribution of phototrophic Dinophyceae to total PE reads varied significantly from 0.4% in May (T9) to 56% in February (T6). Four species of genera *Heterocapsa*, *H. pygmaea*, *H. nei*, *H. rotundata* and *H. triquetra*, were represented by 8 ASVs. *Woloszynskia halophila* was the most diverse (4 ASVs) and abundant group. Moreover, *Ansanella granifera*, *Azadinium trinitatum*, *Biecheleria cincta*, *Gymnodinium dorsalisulcum*, *Karlodinium veneficum* and *Scrippsiella precaria* were found mostly in the February (T6) and March-June samples (T7–T10).

All Haptophyta sequences were represented by Prymnesiophyceae algae, which were the most diverse and abundant in water sample 5976 and accounted for almost 7% of the total PE reads. In the period from September to January (T1–T5), Prymnesiophyceae were not found but occurred at a low abundance (~0.1%) in the February and March samples (T6–T7). In April (T8), Prymnesiophyceae accounted for almost 2% of the PE reads, and its abundance decreased below 1% in subsequent samples. A total six ASVs affiliated to Chrysochromulina (*Ch. acantha*, *Ch. leadbeateri* and *Chrysochromulina* sp.), three ASVs of *Phaeocystis pouchetii*, and four ASVs of unidentified haptophytes were revealed.

Moreover, several ASVs of unidentified Chrysophyceae, Pycnococcaceae, and Chlorophyta were detected at low levels (<1%) in only a few samples (Appendix A).

The Chao1 indices estimated that the diversity of the PE community varied from seven to 46 (Table 1). The highest diversity was found in the March community (L7), and the lowest diversity was found in the September-November communities (L1–L3).

### 3.5. PE Community Structure

The Bray–Curtis dissimilarity result from the PE sequences clustered the samples into four groups (Figure 8). The samples collected in September, October, and November (T1–T3) were grouped together at 60% similarity. The samples collected in December and January (T4 and T5) were clustered together with the May and June samples (T9 and T10) as sister clusters (62% similarity), indicating the minor composition differences in these two groups of samples. The February, March and April samples (T6, T7 and T8) formed the last cluster with 40% similarity. Water sample 5976 was grouped separately as a single branch. SIMPER analysis showed that the dissimilarity between the clusters was determined by the different contributions of total *Micromonas* sequences and other algae groups. Therefore, the T1–T3 cluster was characterized by a high contribution of the genus *Micromonas* to the total number of PE reads (70%), and clusters T4–T5 and T9–T10 was characterized by a high contribution of two Trebouxiophyceae algae, *Picochlorum* sp. (21%) and *Choricystis* sp. (13%), and a contribution of *Micromonas* lower than in the cluster T1–T3 (51%). The similarity of the cluster of the T6–T8 samples was due to *Rhodomonas* sp. (16%) and the moderate *Micromonas* (17%) contribution. Water sample 5976 was characterized by a high contribution from diatoms, particularly *Fragilariopsis cylindrus* (20%).

## 4. Discussion

The samples from the long-term sediment trap allowed us to assess the diversity of small protists (fraction < 3 µm) and provided the comprehensive information on changes in sinking pico-sized communities from September to June in hard-to-reach areas of the Kara Sea. To our knowledge, there is only one study to date in which molecular methods are applied for studying protist community composition in mercury-preserved samples collected with sediment traps [19]. The opportunity to visit the northern part of the Kara Sea in the autumn when the sea is free of ice and deploy the sediment trap for the period until next July made it possible to obtain data for the late autumn, winter and early spring-seasons when this sea area was inaccessible. In our study, we refer to trap samples collected in September, October, and November as autumn samples; samples collected in December, January, and February as winter samples; and samples collected in March, April, May, and June as different spring seasons, i.e., early spring, spring, and later spring, respectively. We understand that such classification may not be entirely accurate since the speed and time of protist sedimentation to a depth of 110 m was not taken into account.

The depth of the sedimentation trap (110 m) was determined by several factors. The depth of the photic layer in the Kara Sea varies considerably over the seasons and can range from 6 to 50 m [39,40]. Thus, setting a trap higher than 50 m is not always justified. Analyses of samples collected by a similar sedimentation trap show that the seasonal succession of large phytoplankton is well detected [41]. Using the data collected by such trap for study of picofraction of phytoplankton is justified and can be an estimate of the development of picoplankton throughout the entire water column, including the surface layer and the photic zone.

The trap was deployed in September when the concentration of Chl_tot_ was low and consistent with Chl_tot_ values found in this region in late summer and autumn [39,42]. The concentration of Chl_pico_ and its relative contribution to Chl_tot_ in the Kara Sea were close to the values of these parameters in September in the Beaufort Sea (0.19 mg/m^3^) [43] and in the Barents Sea (0.52 ± 0.20 mg/m^3^) [44] and slightly higher than those in the central Arctic Ocean (averages of 0.12 mg/m^3^ and 60%) [45]. Such relatively low levels of Chl *a* and high contributions of Chl_pico_ confirm that the sediment trap was deployed in the autumn season. According to Bezzubova et al. [46], the assessment of chlorophyll concentrations in samples fixed with mercury chloride that have been preserved for a long period depends on the taxonomic composition of algae and, thus can lead to incorrect conclusions about the seasonal succession of phytoplankton. In polar waters, the seasonal phytoplankton succession is characterized by a high-biomass spring bloom dominated by centric diatoms. The highest relative read abundance of diatoms in the sediment trap samples occurred in T4 (December) and T7 (March) and was 10% in both samples. In December, all diatom sequences were affiliated with ubiquitous *Ch. socialis*. In earlier studies, the high abundance of *Chaetoceros* spp. was reported in the upper waters north of the New Siberian Islands in October 1893 [47]. In March, the diatom diversity was significantly higher than that in December and included exclusively ice-associated *Melosira arctica* [4,48] and different species of *Thalassiosira*, *Skeletonema*, and *Chaetoceros*, which were the most frequently occurring species during the spring bloom in the Norwegian and Barents seas [49]. The relative abundance of the main picophytoplankton representative, Mamiellophyceae, was two times lower in March (T7) than in April (T8) and January (T5). In earlier studies, a negative correlation between the diatom spring blooms and the picoeukaryote diversity was found in the Arctic [50] and in the English Channel [51]. Thus, it can be assumed that the March sample was from an under ice bloom in the Kara Sea.

In line with other studies, all samples filtered through a 3-μm pore size filter contained a significant number of sequences corresponding to protists with cells ≥3 μm [32,52,53]. One of the reasons for this result might have been the use of a filter with a 0.2 µm pore size. The extracellular DNA (particulate or dissolved) of large nano- and microplanktonic cells in water was retained onto 0.2 µm filters, through a collection of aggregates or molecular adsorption [54]. Moreover, although low vacuum pressure was applied during sample fractionation and filtration to minimize cell breakage or disruption of cells in chains, fragile cell groups, such as naked dinoflagellates and ciliates, could have been destroyed. All the above factors increased the number of non-pico-sized protist reads in the samples.

Surprisingly, the taxonomic composition of microbial eukaryotes in water sample 5976 significantly differed from that in sediment trap sample T1, although both were collected in September one after the another. The water sample was characterized by a high abundance of Ciliophora reads, while trap sample T1 contained an order of magnitude fewer Ciliophora sequences. It should be noted that ciliates were found in low abundance in all trap samples. Although ciliates range from 10 μm to a few millimeters in cell size, they are repeatedly reported in the pico-sized fraction (smaller than 2–3 μm) of molecular surveys [50,54,55,56,57]. Ciliates are ubiquitous in marine plankton at a global scale [58,59] and play an important role in the food web, but only 20% of determined ASVs were identified at the genus level. To date, Ciliophora has seldom been studied in metabarcoding analysis; and thus represents a promising additional resource for further study. Differences in the abundance of Ciliophora reads between the trap samples and the water sample can be explained firstly by local ciliates blooming in September where the trap was set and secondly the destruction of naked cells and their utilization by heterotrophs during long-term storage in sediment trap cups.

Fungi are the second group that distinguished the diversity of September water sample 5976 from the of all the sediment trap samples. The high contribution of fungal reads in the sediment trap communities (10–42%) indicates that the mercuric chloride used for preserving the samples during deployment to inhibit of bacterial activity has no effect on Fungi. Different *Rhodotorula* species, in particular *Rhodotorula mucilaginosa*, which belongs to the phylum *Basidiomycota*, were common and numerous in all trap samples. Earlier Liu et al. (2017) [60] isolated and identified a mercury-tolerant yeast isolate of *Rhodotorula mucilaginosa* R1. This isolate was able to grow in the presence of a high concentration of Hg^2+^. According to these data, it could be deduced that the high fungal abundance in the trap samples was explained by the presence of such Hg-tolerant fungal species. Within Fungi, Basidiomycota and Ascomycota dominated, while Cryptomycota and Microsporidiomycota were identified at low abundances. Previously, Basidiomycota and Ascomycota were detected in the seawater at the North Pole and in the Baltic Sea [61,62]. Marine Fungi are well known to be as heterotrophs that play an essential role as decomposers of organic matter [63,64], and this role was confirmed by the obtained results regarding their high abundance in the trap samples and almost complete absence in the water sample. Fungal diversity in our samples suggested their importance in the processing of organic matter in the Arctic open-sea ecosystem [65,66].

As mentioned above, Dinoflagellata was the third most abundant group in all the studied samples. Its contribution to total read abundance in water sample 5976 was comparable to that in the sediment trap samples from January to May (T5–T9). The dominance of Dinoflagellata in our samples could be explained mostly by the abundance and diversity of Syndiniophyceae–a diverse parasitic group common in all marine environments [67,68,69]. This group can infect a variety of hosts, from marine animals to protists, although many of them are specific parasites of other dinoflagellates [70]. The presence of Syndiniophyceae was observed in North Pole sea ice and water samples [61], and in other regions of the Arctic Ocean [71,72]. Our results indicate that Syndiniophyceae are key players in polar waters in all seasons, but currently, its taxonomic diversity and ecological role remain poorly understood. Phototrophic Dinophyceae were most diverse in the late winter and spring samples (T6–T9) when the length of daylight hours increased. The revealed high relative abundance of Dinoflagellata in the winter samples corresponded to the assumption of [61] that dinoflagellates may be diverse and abundant at the end of the polar night in both sea ice and seawater.

The dominance of phototrophic algae in late autumn was an unexpected result of our study. The detailed analysis of PE communities showed that Mamiellophyceae were the predominant PEs in most studied seasons. In polar water ecosystems, Mamiellophyceae are one of the prevalent classes of photosynthetic picophytoplankton [13,73,74,75]; their cell sizes varied from 0.8 to 3.0 μm [68,76,77]. The widespread abundance of *M. polaris* in all trap samples indicated that it is an important picoplanktonic primary producer in all seasons. This tiny alga can retain its pigments and survive throughout winter [13]. In all the autumn samples (T1–T3), *M. polaris* represented a major constituent of phototrophic PEs due to its fast growth rates under low light and low temperatures.

The high abundance of Trebouxiophyceae, especially in winter and early spring samples, indicates that these algae play no smaller role than that of Mamiellophyceae. Trebouxiophyceae are among the most common terrestrial and freshwater algae in polar regions. The most abundant *Choricystis* sp. and *Picochlorum* sp. found in our samples include genera belonging to freshwater, brackish water and marine species. Considering that the surface layer of the studied area is influenced by river runoff [78], it can be assumed that representatives of both genera may belong to freshwater or brackish water species. It has been found that multiple *Picochlorum* isolates can adapt to rapidly changing environments and expand habitat range from mesophilic to halophilic through allelic diversity, with minor but important contributions from horizontal gene transfer [79]. Some Trebouxiophyceae, e.g., *Botryococcus* cf. *braunii*, were detected in the Kara Sea sediments over 100 to 500 km offshore, their persistent occurrence in palynomorphs assemblages was related to the discharge of freshwater from the large Siberian rivers into the Arctic shelf seas [80]. In September 2017, Trebouxiophyceae were identified in the water of the Kara Sea at a low abundance (<1%) [27], but in this study, Trbouxiophyceae were not found in September in water sample 5976.

Cryptophyta were regularly part of the PE communities during the studied seasons, but in the early spring samples (T7–T8), their contribution increased and exceeded 10%. Such results agree with the data obtained from the North Sea, where pigmented cryptophytes contributed throughout the whole year to the phytoplankton community, with peaks observed in early spring and late summer [81]. In the Arctic, when sea ice melts and nutrients are released to the open water column, Cryptophyta are also part of the spring bloom [48,82,83,84]. The abundance of Cryptophyta ASVs in the studied samples may be explained by the methodological considerations because photosynthetic pico-sized representatives of Cryptophyta are not currently described, with the exception of the cryptophyte *Hillea marina* [77]. Pico-sized cryptophytes have been identified with FISH probes in the Baltic Sea [85]. Moreover, in the process of sexual reproduction of some Cryptophyta, they had divided by meiosis into smaller celled gametes, which have been collected in the picoplankton fraction [86].

We found a relatively low abundance of Dictyochophycea, Bolidophyceae, and Haptophyta in all the trap samples in comparison to that in water sample 5976. It is currently difficult to conclude whether the differences in abundances of these algal groups reflect a methodological bias associated with combining molecular methods and long-term preservation by mercury chloride or whether the composition of the picoalgal community has such a seasonal dynamic. Data on the taxonomic composition and seasonal succession of different algae in low-latitude open waters in autumn and winter are limited and absent in the Russian Arctic seas. The obtained results characterize the seasonal dynamic of picophytoplankton community in the northwestern part of the Kara Sea and will serve as a baseline for ongoing research of diversity and seasonal plankton dynamics, not only in the Kara Sea but in other seas of Russian Arctic.

## 5. Conclusions

The present study is the first to report on the seasonal cycle of photosynthetic microbial eukaryote (<3 µm) diversity, using a 10-month sediment trap record (2018–2019), deployed in the northern area of the Kara Sea. Rotating sample cups allowed us to analyze the taxonomic composition sequentially from month to month from September to June. However, it turned out that the use of the long-term sediment traps has its limitations caused by mercury chloride preservation solution, which did not influence Fungi. Unexpectedly, in late autumn, photosynthetic algae dominated in terms of total protist read abundance, but their diversity was lower than that in other seasons. Generally, our data indicate that communities of phototrophic eukaryotes filtered through a 3 µm pore size are diverse and consist of at least 15 algal classes throughout the three studied seasons. Mamiellophyceae, in particular, *Micromonas polaris*, was constantly present throughout the September–June period in the PE community, with dominance in the autumn, winter, and late spring seasons. In early spring, during the under-ice diatom bloom, the relative contribution of Mamiellophyceae to the total PE abundance decreased. Our results show that in areas influenced by river runoff, Treboxiophyceae can play a significant role in phytoplankton during low-light conditions. Cryptophyta, Dictyochophycea, Bolidophyceae, and Haptophyta were also present, albeit their abundance was low. Long-term studies of phytoplankton seasonal succession in low-latitude waters are minimal, and the application of molecular methods has become an important tool for comprehensive taxonomic analyses of protist communities in long-term sediment traps but with some limitations.

## Figures and Tables

**Figure 1 plants-10-02394-f001:**
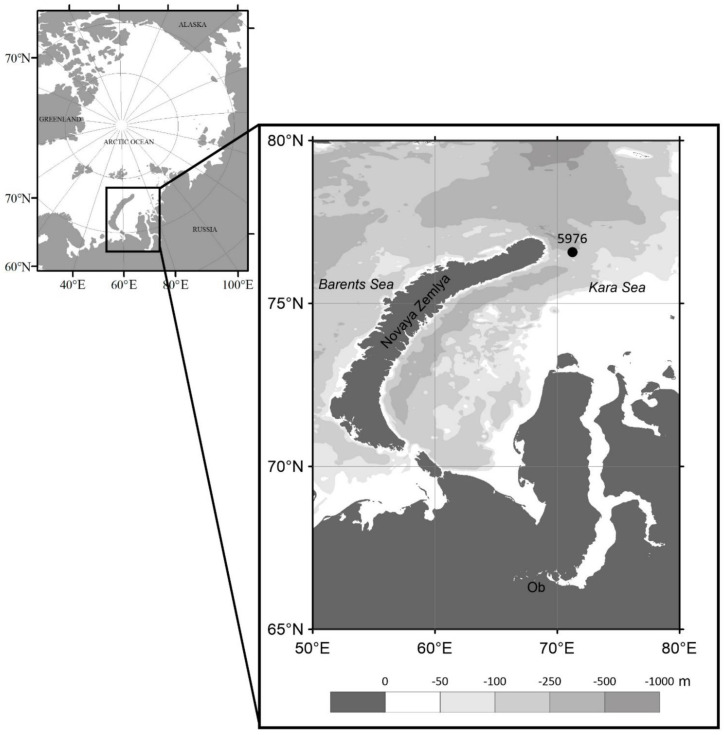
Bathymetric map of the Kara Sea with the location of the moored sediment trap at station 5976.

**Figure 2 plants-10-02394-f002:**
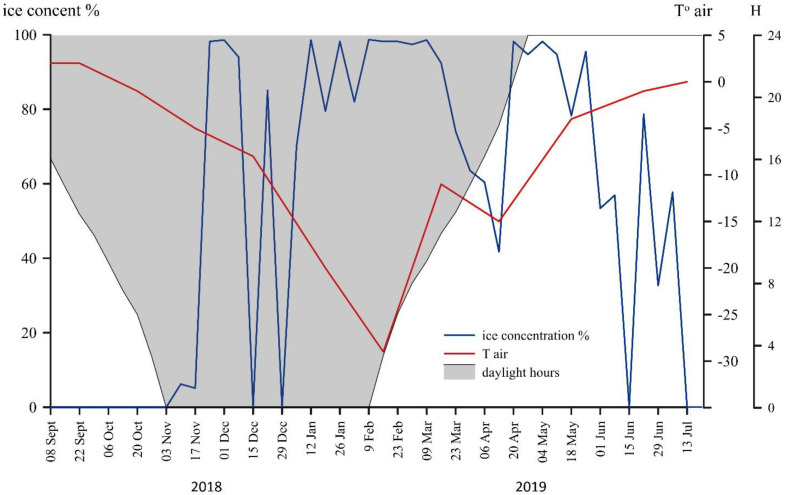
Annual course of air temperature and ice cover (based on data from http://www.aari.ru, accessed on 1 September 2021) in the area of the studied station 5976, the Kara Sea.

**Figure 3 plants-10-02394-f003:**
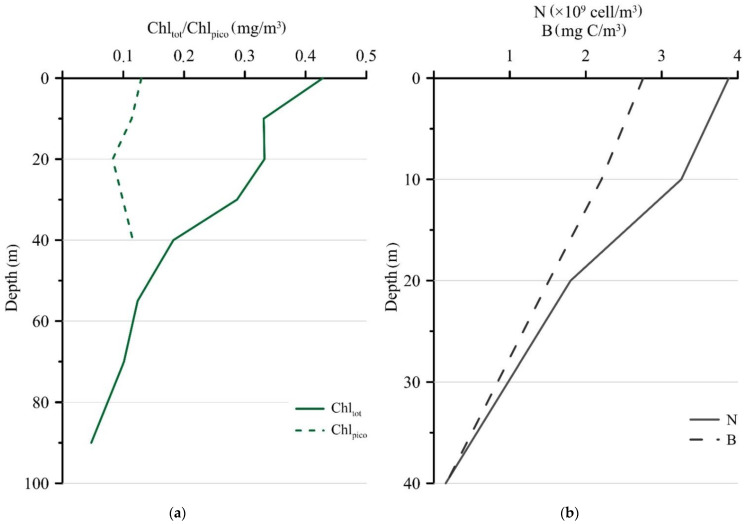
Vertical distribution of Chl_tot_ and Chl_pico_ (mg/m^3^) (**a**), photosynthetic picoeukaryotes abundance (N, ×10^9^ cells/m^3^) and biomass (B, mg C/m^3^) (**b**) at station 5976 during the setting of sediment trap in September 2018.

**Figure 4 plants-10-02394-f004:**
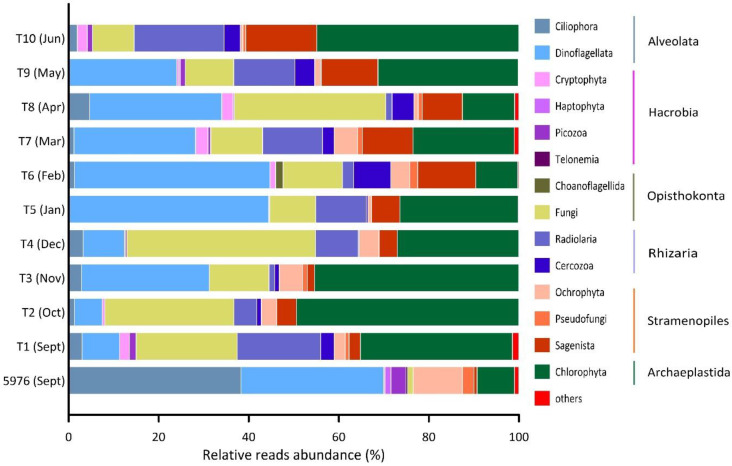
Relative reads abundance (%) of the major taxonomic groups in the studied samples. “Others” includes groups with the contribution <1% (Apicomplexa, Perkinsea, Conosa, Centroheliozoa, Mesomycetozoa, Opalozoa).

**Figure 5 plants-10-02394-f005:**
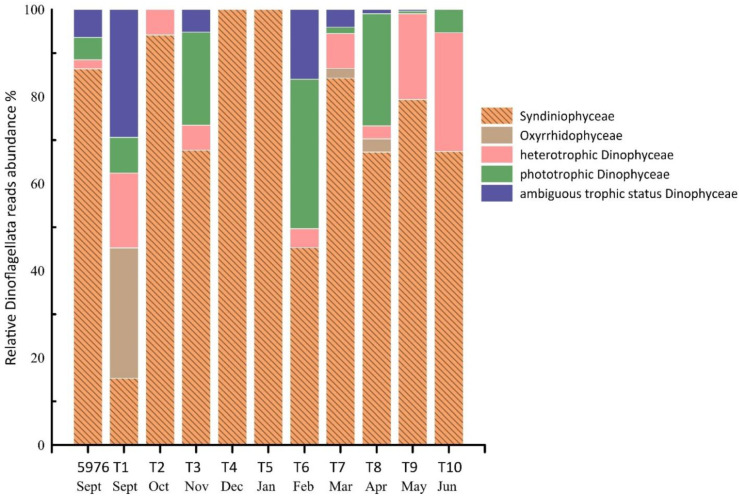
The contribution of Syndiniophyceae, Oxyrrhidophyceae, and Dinophyceae (%) to the total Dinoflagellata reads in the studied samples.

**Figure 6 plants-10-02394-f006:**
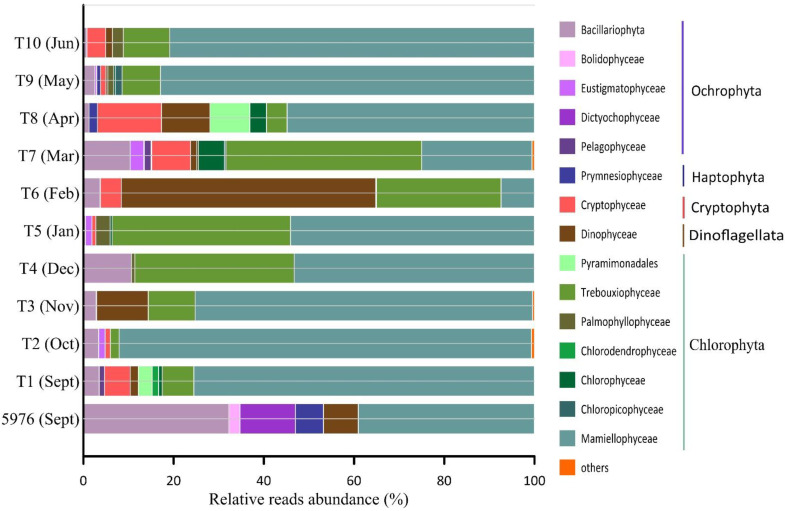
Relative reads abundance (%) of different PE groups in the studied samples. “Others” includes groups with the contribution <1% (Pycnococcaceae_X_sp., Prasino-Clades, Chrysophyceae).

**Figure 7 plants-10-02394-f007:**
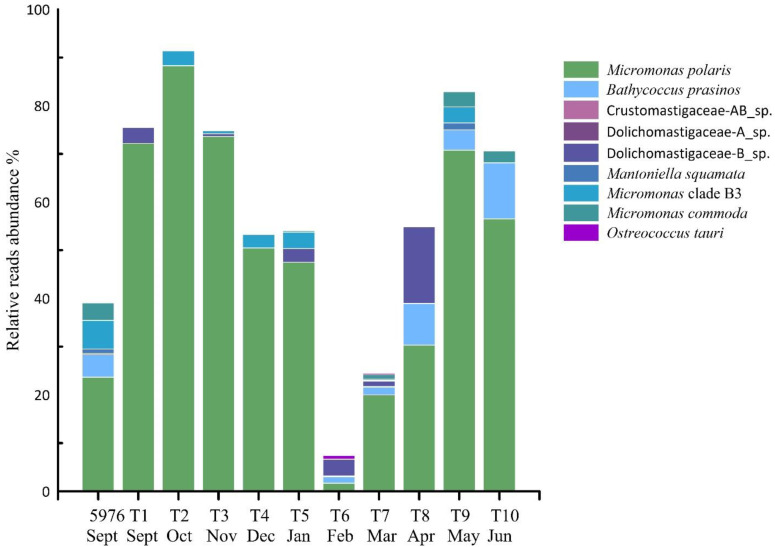
The contribution (%) of Mamiellophyceae phylotypes to total PE reads in studied samples.

**Figure 8 plants-10-02394-f008:**
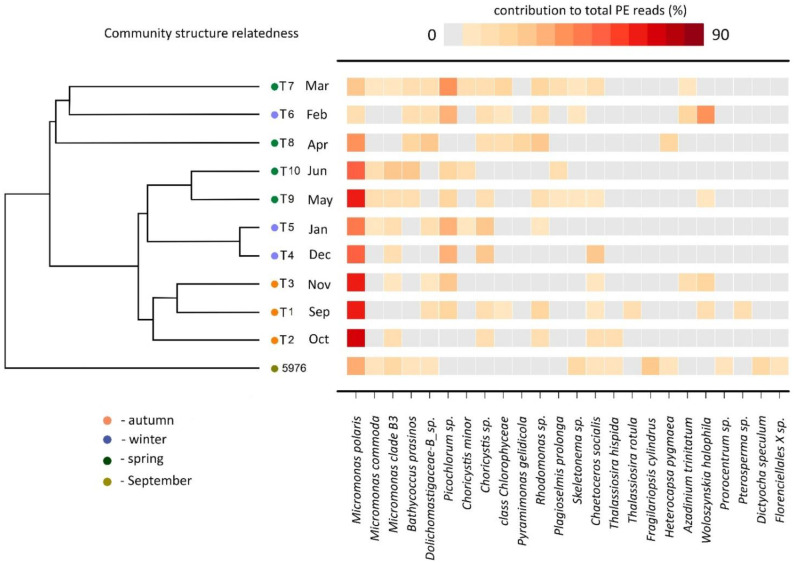
Heatmap showing the V4 reads representation of the 24 most abundant phototrophic eukaryotes. Community structure related cluster dendrogram was calculated using Bray–Curtis similarity based on the square root transformed relative reads abundance for each sample.

**Table 1 plants-10-02394-t001:** Sequencing statistics and estimated PE ASVs richness and diversity index Chao 1 obtained from metabarcoding analysis of one water sample and sediment trap samples in the Kara Sea picoeukaryote communities.

Sample	Date/ Period	Total No. of V4 Tag Quality Sequences	Quality Sequences	No. of Protist Reads	No. of Protist ASVs	No. of Photosynthetic Eukaryotes Reads	No. of Photosynthetic Eukaryotes ASVs	Chao1
5976	8 September	304,129	129,900	127,795	89	27,164	28	22
T1	September	157,252	118,966	94,543	62	36,344	16	13
T2	October	110,720	86,035	74,141	47	38,955	13	11
T3	November	202,474	160,279	139,922	66	74,172	19	14
T4	December	127,790	93,930	71,426	45	21,589	13	7
T5	January	260,009	217,018	177,565	71	47,926	22	15
T6	February	190,536	146,055	125,063	106	33,073	31	19
T7	March	236,408	153,141	133,153	257	40,360	81	46
T8	April	221,949	145,227	94,280	85	15,347	21	13
T9	May	232,284	159,887	137,194	169	45,242	56	26
T10	June	385,572	105,569	76,089	55	30,434	17	10

## Data Availability

The data generated or analyzed during this study are included in this published article and its Appendix A.

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
