# Peer review of "Seasonal Variability of Photosynthetic Microbial Eukaryotes (<3 µm) in the Kara Sea Revealed by 18S rDNA Metabarcoding of Sediment Trap Fluxes"

_plants, 2021, doi:10.3390/plants10112394_

Round 1

Reviewer 1 Report

The manuscript authored by Belevich and coauthors investigate the photosynthetic eukaryotes in terms of biodiversity and seasonality in Kara Sea with sediment trap fluxes approach.

I believe that the manuscript well written, and the results are relevant for the scientific community.

I found the study interesting as it brings new insights into the dynamics of such organisms in polar areas.

I have few comments, reported below, but I think the manuscript can be accepted after minor revisions.

Figure 1: I think another map nearby the one presented with a larger overview of the area will help localize the study site.

L 122: Chl A should be corrected to Chl a.

L 144:150: What was the sequencing read length?

L 392-395: Here and in Table1 you present the result of alpha diversity index (Chao1), but these are not described in M&M, so these information should be added.

Figure 8: Why the samples here are named “L” instead of “T”? Also, if you report the month near the sample name it will make the figure more readable.

Author Response

We are extremely grateful to the Reviewer for the careful analysis of the manuscript and valuable comments which allowed us to improve the text of the article. We have prepared a revised version of the paper taking into account the comments by the Reviewer.

Reply to Reviewer #1

Comment 1. Figure 1: I think another map nearby the one presented with a larger overview of the area will help localize the study site.

Response: Figure 1 was modified; map of the Arctic region has been added in the revised version.

Comment 2. L 122: Chl A should be corrected to Chl a.

Response: It has been changed in the revised version.

Comment 3. L 144:150: What was the sequencing read length?

Response: The read length was 250 bp; reading was performed from both sides of the fragment. It has been added to Mat&Meth section in the revised version.

Comment 4. L 392-395: Here and in Table1 you present the result of alpha diversity index (Chao1), but these are not described in M&M, so this information should be added.

Response: All diversity indices were calculated using Primer v6 software. It was written in section Statistical analyses of Mat&Meth (L 166-168). The clarification was added to the text.

Comment 5. Figure 8: Why the samples here are named “L” instead of “T”? Also, if you report the month near the sample name it will make the figure more readable.

Response: The samples names were changed and the month near the sample name were added to the revised version.

Reviewer 2 Report

The manuscript of Belevich refers to a NGS approach to study photosynthetic microbial eukaryotes (<3μm) in the Kara Sea. Though the information presented is of relevance the authors lack some clarifications. First why was this study undertaken and in this region and specifically in the sampling point? What is the comparison to other NGS data on other polar regions? Why the need to study this group of organisms? I mean where they first recorded in this study or they were previously found or studied in the region? What are the ecological impacts of the taxa detected in this region?

Another remark is what is the major contribution of their work? By sampling one point do they know the taxa profile in this region?

Another question is about the limitations of the experiments. How can they surpass them?

Author Response

We are extremely grateful to the Reviewer for the careful analysis of the manuscript and valuable comments, which allowed us to improve the text of the article. We have prepared a revised version of the paper taking into account the comments.

 Reply to Reviewer #2:

Comment 1. First why was this study undertaken and in this region and specifically in the sampling point? What is the comparison to other NGS data on other polar regions?

Response: The Kara Sea belongs to the Arctic seas, and studies of picophytoplankton diversity in this region have been carried out only in early (end of August) and late autumn (end of September) (Belevich et al., 2020). Complex studies of carbon fluxes and the role of river runoff in sedimentation in the Kara Sea have been carried out for over 10 years. Earlier, sedimentation traps were installed on the shelf of the Kara Sea in the area of high-impact river runoff. Station 5976, located northwest of the Kara Sea, was selected to assess the fluxes in the area with the low impact of river runoff. Thus, our studies of the species composition of plankton picofraction became part of the ongoing studies in the Kara Sea. In the Introduction, we present studies where phototrophic picoeukaryote diversity with the application of molecular methods has been studied in the Arctic Ocean (Kilias et al., 2014; Metfies et al., 2016; Meshram et al., 2017; Kirkham et al., 2013). These studies were carried out in the spring and summer seasons, while our data allow us to found the species composition and its dynamics during the period from autumn to spring during polar night and ice-covered period. In Discussion we comparison our data with data on other polar regions where it was possible (e.g. L 559-560, 572, 591).

Comment 2. Why the need to study this group of organisms? I mean where they first recorded in this study or they were previously found or studied in the region? What are the ecological impacts of the taxa detected in this region?

Response: In Introduction, we present previous studies where photosynthetic picoeukaryotes are one of the major components of phytoplankton in the Arctic region – these tiny algae are responsible for significant photosynthetic production and therefore have a great impact on marine carbon and energy budgets (Gosselin et al., 1997; Sherr et al., 2003; Bravya et al., 2018). Earlier picophytoplankton studies in the Kara Sea have been carried out only in autumn when the sea is free of ice (Belevich et al., 2020, 2021). In Discussion, we underline that most of the identified taxa were not previously recorded in the Kara Sea, but they are present in the communities of picophytoplankton in the Arctic Ocean.

Comment 3. Another remark is what is the major contribution of their work? By sampling one point do they know the taxa profile in this region?

Response: Our results show the taxa profile and seasonal dynamics of picoplankton in the autumn-winter period of 2018-2019 in the northwest part of the Kara Sea. The hydrochemical and hydrophysical conditions in the Kara Sea very significantly due to interaction of freshwater runoff, Arctic basin waters, and transformed Atlantic waters. Our results obtained only for one area of studied sea, they are pilot, we plan to continue our studies to assess the temporal dynamics of picoplankton communities in the rest of the Kara Sea, as well as in other seas of the Russian Arctic - the Laptev and East Siberian seas. We hope that the work will be of interest for specialists and can serve as a basis for further studies of picophytoplankton dynamic in the Arctic seas.

It was added in the revised version: “The obtained results characterize the seasonal dynamic of picophytoplankton community in the northwestern part of the Kara Sea and will serve as a baseline for ongoing research of diversity and seasonal plankton dynamics not only in the Kara Sea but also in the other seas of Russian Arctic”.

Comment 4. Another question is about the limitations of the experiments. How can they surpass them?

Response: The polar regions, especially the regions of the Arctic seas, are hard to reach areas. The polar night period and ice conditions do not allow conducting research all year round. Therefore, the installation of such a trap is the only way to obtain data on the species composition of picophytoplankton in the autumn-winter period. Setting up several annual traps at the same time in the same area is too expensive. We plan to conduct a similar study in other areas of the Kara Sea and repeat the experiment in this area to clarify the obtained results and assess the impact of environmental factors in different areas of the sea on the species composition of picophytoplankton.

Reviewer 3 Report

This manuscript by Belevich et al. describes the taxonomic composition of microbial eukaryotes in the Kara Sea as determined using a sediment trap and NGS 18S rDNA metabarcoding. They found seasonal changes in phototrophic eukaryotes that enhance our understanding of polar maritime ecosystems.

Generally, I found this manuscript well-written, detailed, and informative. It will provide a valuable resource for those interested in pelagic eukaryotic microbial diversity. The data appears to be of excellent quality, and their analyses reasonable. Inclusion of primary data (fasta sequences) in Supplementary Materials is welcome. I especially enjoyed their thoughtful Discussion, which indicated a deep understanding of their system and its limitations. For the most part, this manuscript is ready for publication in a journal such as Plants. However, I have a few comments that, if addressed, may help to improve it.

[Figure 1] While a map indicating the location of the sediment trap is most necessary, I would have preferred a picture of a sediment trap as the first figure. Many readers might be unfamiliar with this device (surface buoyed, moored, neutral buoyancy?), and it does play a crucial role in this research.

[Figure 4] Data are provided as relative abundance, which is informative. However, absolute abundance would be even better. Can this be determined from your data? It would be illustrative to observe the changes in numbers of organisms as well as their taxonomic groups throughout the seasons at this point in the paper.

[Figure 6] I do not understand the existing Figure 6. It looks like a repeat of Figure 5. Please correct.

[Data Availability Statement] I could not find the original data on Figshare. Perhaps these could be included with the Supplementary Materials hosted by the journal Plants?

Author Response

We are extremely grateful to the Reviewer for the careful analysis of the manuscript and valuable comments, which allowed us to improve the text of the article. We have prepared a revised version of the paper taking into account the comments.

 Reply to Reviewer #3

Comment 1. [Figure 1] While a map indicating the location of the sediment trap is most necessary, I would have preferred a picture of a sediment trap as the first figure. Many readers might be unfamiliar with this device (surface buoyed, moored, neutral buoyancy?), and it does play a crucial role in this research.

Response: A image of a sediment trap is given in the Graphical Abstract that located alongside the text abstract.

Comment 2. [Figure 4] Data are provided as relative abundance, which is informative. However, absolute abundance would be even better. Can this be determined from your data? It would be illustrative to observe the changes in numbers of organisms as well as their taxonomic groups throughout the seasons at this point in the paper.

Response: The relative contribution of reads usually is used to interpret the results in most studies devoted to the species composition of planktonic protists with application of NGS method (e.g. Tragin et al., 2017 doi:10.1111/1462-2920.13952; Kilias et al., 2013 doi:10.1111/jpy.12109; Monier et al., 2016 doi:10.1111/1758-2229.12390). The data of absolute reads abundance do not always correctly show the role of certain organisms in the community, since the total number of reads in each sample can differ significantly. For example, in our work, the number of reads varies by more than 3 times - from 127790 (sample T1) to 385572 (sample T10). However, the absolute reads abundance for each taxon are presented in Table S1, included in Supplementary Materials hosted by the journal Plants.

Comment 3. [Figure 6] I do not understand the existing Figure 6. It looks like a repeat of Figure 5. Please correct.

Response: We are very sorry the Figure 6 was wrong. The Figure has been replaced with the correct one.

Comment 4. [Data Availability Statement] I could not find the original data on Figshare. Perhaps these could be included with the Supplementary Materials hosted by the journal Plants?

Response: All Supplementary Materials are attached to the manuscript.

Round 2

Reviewer 2 Report

The authors have addressed my comments and the manuscript is now suitable for publication.